# Barriers and Facilitators of Working with Dying Patients among Novice Nurses in Saudi Arabia

**DOI:** 10.3390/healthcare10112259

**Published:** 2022-11-11

**Authors:** Turki S. Alsalamah, Yasir S. Alsalamah, Basmah Aldrees, Thamer Alslamah, Sarah M. Yousif, Mirna Fawaz

**Affiliations:** 1MSN, RN, Department of Nursing, Clinical Simulation, King Fahed Specialist Hospital, Ministry of Health, Buraydah 52367, Saudi Arabia; 2Department of Nursing, Mental Health Hospital, Ministry of Health, Buraydah 52367, Saudi Arabia; 3College of Nursing, King Saud Bin Abdulaziz for Health and Sciences, Riyadh 14611, Saudi Arabia; 4Department of Public Health, College of Public Health and Health Informatics, Qassim University, Al Bukairiyah 52571, Saudi Arabia; 5Department of Nursing, Critical Care, King Abdulaziz Medical City, Ministry of National Guard-Health Affairs, Riyadh 11426, Saudi Arabia; 6Faculty of Health Sciences, Beirut Arab University, Beirut 1105, Lebanon

**Keywords:** end of life, death, terminal care, barriers, facilitators

## Abstract

Novice nurses face immense challenges while they transition from being students to becoming professional nurses. Dealing with dying patients has been documented to be an immense task among professional nurses, especially for new nurses. This study aimed to explore the barriers and facilitators of working with dying patients, experienced by novice nurses in Saudi Arabia. This study employed a phenomenological qualitative research methodology among the nurses who participated in this study, which were twelve participants, of various ages, genders, religions, and nationalities, who took part in semi-structured focus group discussions. The identified barriers included a lack of experience with dying patients; a language barrier with patients and medical staff; inadequate staffing; and patients’ responses or cooperation. Furthermore, the facilitators were categorized into three themes: caring/compassion, teamwork/collaboration, and mentors/experienced colleagues. Based on the findings of this study, an action plan must be developed to improve the experience of novice nurses in Saudi Arabia, and to minimize the impact of the barriers on the new nurses when working with such patients.

## 1. Introduction

Dying is a painful time in an ill person’s journey; it causes concern for not only the terminal patient but also others surrounding them. Nurses operate in an atmosphere where death and dying are frequent. Every mortality is a traumatic experience, yet it is inextricably rooted in human presence. Nurses are frequently present with a dying patient and are expected to perform their jobs professionally while coping with such a client. Such scenarios are clearly associated with intense sentiments and substantial anxiety for nursing personnel [1,2].

Work fatigue amongst nurses is mostly caused by the strain of tending to dying clients. Long-term care professionals experience work fatigue as a result of their feelings related to the patient’s mortality [3]. The loss of a patient is one of the most distressing aspects of a nurse’s job, second only to a lack of personnel and supplies. Prior research looked at healthcare teams’ dread of death and their strategies of dealing with it in various situations. Medical personnel’s dread of clients’ death is prevalent, and it is linked to a poor mindset regarding care for a dying patient [4].

Mental endurance and subsequent traumatizing strain in nurses dealing with dying patients were studied by Ogiska-Bulik and Michalska. The findings suggest that, aside from interactions with the client and his or her relatives, disagreements with supervisors, and ambiguity about the treatment implications, the mortality of patients generated massive strain, far larger than that encountered in other elements of the profession [5].

There is not much information in the previous research about the sentiments that nurses experience while dealing with a dying client. Furthermore, only a few papers tackle coping techniques for those dealing with death and dying patients. Caring for a dying person in the final minutes of his or her life is challenging, which contributes to the high anxiety level connected to the devastating situation. The capacity to find oneself in a scenario where a patient dies, the necessity to help the patient’s relatives, and the necessity to cope with one’s own feelings are all aspects that impact nurses’ behavior as health professionals and as fellow humans. In the current state of the healthcare system, nurse practitioners are among the most stressed professionals [6].

Shift work, which disturbs the normal sleep cycle; excessive physiological and psychological burden; organizational issues at the workplace; interpersonal communication challenges; unsatisfying compensation; uncertain professional growth paths; job instability, and many other factors lead to anxiety. Nurses who operate in places where they come into contact with dying patients believe that death causes intense emotions and tension [7]. Generally, one of the crucial circumstances in nursing is when a patient dies. Despite their capacity to maintain self-control and a peaceful attitude in the situation, the nurses must undoubtedly “process out” certain feelings. The most prevalent are feelings of powerlessness and desertion, as well as anger and sadness. Emotional distress causes a nurse to minimize interactions with the dying individual and his or her family members. The death of a patient has an impact on the nurse’s career as well as personal life, perhaps leading to a loss of identity and a sense of shame [8]. Therefore, this study was conducted to explore the barriers and facilitators of working with dying patients experienced by novice nurses in Saudi Arabia.

## 2. Materials and Methods

### 2.1. Research Design

The purpose of this study is to gain a deeper understanding of novice nurses’ encounters while tending to dying patients using a phenomenological exploratory analytical method. The method used in this research is founded upon Colaizzi’s phenomenology model [9], which uses respondents’ viewpoints and observations to describe the event under investigation, leading to the finding of common rather than individualized traits among the sample group [10]. The seven phases of Colaizzi’s technique were employed in this research to help establish relevant themes. This strategy allowed the investigators to gain a deep grasp of the nurses’ viewpoints, as well as properly explore the data in order to uncover meaningful themes. This study examined the respondents’ perspectives using a constructivist epistemology, keeping in mind that they are situational, personal, and impacted by how people perceive the world and react to experiences [11].

### 2.2. Setting and Sample

This study used a purposive sample of 12 new nurses at the emergency department of one of the most advanced hospitals in Buraydah, Saudi Arabia, which was established in 1981. It has a large ER department, and it is considered one of the largest hospitals in Saudi Arabia. Moreover, it covers more than 574 beds that are distributed across all of the hospital’s units. Moreover, its emergency department offers helicopter-landing sites to accommodate any emergency situation, especially in case of natural disasters. In total, 12 of the 16 eligible nurses replied to the request to enroll in the research, resulting in a 75 percent response rate. When questioned as to why they did not partake in the research, the remaining four nurses stated their schedule did not permit their participation. The interviewees had at least three years or less of working experience with caring for dying patients or sudden patients’ deaths. The participants volunteered to be a part of the study. Nurses who had previously worked in a separate profession or on a different unit were omitted from the research. Nurses with the most diversity were chosen in order to obtain in-depth and complete data (age, gender, and nationality). The sample size necessary for the phenomenological research’s focus group discussion was determined using Krueger and Casey’s (2015) recommendations, which stated that five to eight people for each meeting were adequate. The goal and technique of the focus group were carefully described to the novice nurses at the approached institution, and they joined the group. Two focus groups with a total of six nurses were conducted [12].

### 2.3. Recruitment and Data Collection

After researchers were granted access to a mail registry, nurses were contacted through email. Nurses were emailed a typical invitation to join the study, along with a summary of the study’s aims; if they accepted, they were requested to sign an informed consent form. Respondents were asked to engage in semi-structured focus group sessions through a virtual meeting platform after an orientation procedure. Data collection was extended from 4 February to 21 February 2022 once the required sample was recruited.

### 2.4. Interviews

A post-graduate scholar and two PhD-holding researchers, who worked as assistant professors at the institution, led the focus group sessions. Two male and one female investigators, each of whom had prior expertise with qualitative evaluation, interviewed the nurses. The investigators had no prior contact with the nurses who were questioned. After receiving the email requesting them to join, the nurses were linked to the investigators by mutual coworkers. The researchers used virtual meeting technologies to perform the discussions, which enabled them to hold discussions until data saturation was reached. A suitable timeframe for the discussions was decided upon with the participants in order for them to be available and provide accurate reports of their encounters, especially considering their heavy workload. The investigators alternated the conversations to prevent the potential of a moderator’s predominance. Each group discussion session ran for between 35 and 45 min, and there were no duplicate interviews. Two groups of respondents were formed. Because the positive group dynamic promoted information exchange, focus group conversations with peers increased engagement [13]. Observation records were taken during and after the interviews to document the researchers’ observations of nonverbal signs, which helped with data processing. The following questions were asked, where the interview guide was based on Krueger and Casey (2015)’s questioning manual (Table 1).

### 2.5. Data Analysis

Each conversation was transcribed and printed in its entirety. Each text was compared with its audio to ensure exactness. To acquire a clear comprehension of the content, the investigators combed over the texts and listened to each audio file numerous times. As per the Colaizzi approach, the researchers read the data repeatedly, extracted significant data, coded recurring data, and lastly organized the data as themes. The Colaizzi method consists of seven steps: (1) read the transcript to be familiar with the data; (2) identify and extract significant statements and phrases; (3) formulate meanings; (4) group all formulated meanings into categories, clusters of themes, and themes; (5) define all emergent themes into an exhaustive description; (6) describe the fundamental concepts of the phenomena; and (7) approach participants for further information [14]. Since the discussions were performed in Arabic, the interview transcripts were transferred to two subject-matter experts who conducted translation and back-translation, before forwarding the transcripts to an independent specialist for verification. The transcripts were coded and anonymized, with each participant being given a pseudonym. The transcribed texts were provided to the translation specialists in a closed envelope, and they were returned in a closed envelope. All researchers completed their separate analysis before meeting and discussing their results, until they reached an agreement on the emerging themes. The quotations were given a descriptive and insightful explanation that captured the essence of the data, and the phrases were then merged, restructured, and compiled into qualitative themes that the investigators hoped would provide a precise and full explanation of the nurses’ experience [15].

### 2.6. Trustworthiness and Credibility

The researchers used a variety of approaches in accordance with past qualitative research to increase the study’s credibility and prevent biases from emerging [16]. Because of the researchers’ extensive involvement with the data and effective interactions with the participants, the results have garnered confidence. As a result, the researchers worked on the project for around two months. Members checked in to assess for consistency between data-driven conceptions and participant views. An expert in qualitative research evaluated replicability. The retrieved codes and themes were sent to peers for external evaluation, and their suitability was investigated and verified. Additionally, maximal variance sampling helped with data generalization and validity. To ensure reliability, the researchers documented and made public the whole research proposal, allowing others to undertake additional research. Concurrent interviews allowed the assumptions to be completely investigated using simultaneous data analysis, culminating in a thorough understanding of the occurrences. All researchers employed the same questioning techniques and asked the same questions, making sure to properly examine any new ideas to eliminate any blind spots in the results. To synthesize the research findings, several phrases were employed, providing the research participants with a distinct voice [17].

### 2.7. Ethical Considerations

The university’s Research and Ethics Committee gave the researchers permission to proceed (ECO-R-160). The purpose of this study was to obtain a profound understanding of the facilitators and impediments of working with dying patients among novice nurses. Respondents had already been briefed about the study’s goals and had signed a formal informed consent declaration to participate. The participants gave permission for the virtual discussions to be recorded. The recordings were kept in a folder that was password-locked. This project was conducted in conformity with the ideals and standards of the International Declaration of Helsinki.

## 3. Results

The sample of this study was made up of 12 novice nurses from one major hospital in Saudi Arabia, where eight (48.57%) of them were males while four (51.43%) were females. The age of the participants ranged from 25 to 35, where five (41.6%) of them were between 26 and 30 years of age. Five of the participants (41.6%) were Saudi, while seven (58.3%) were Filipino (Table 2).

An opening question was asked at the beginning of each interview to assess each participant’s perception regarding being a novice nurse. Most of their answers included their feelings of making a difference in the nursing society in Saudi Arabia. One participant had described it as follows: “since I was a kid, I had a passion for helping people who are in need.” Another one described it as follows: “I like taking care of the patient who needs our help.” Another one had said, “I love to care to other people.” Moreover, all participants described the environment in Saudi Arabia as unique because it has a lot of diversity; thus, nurses are able to interact with people who are from different cultures, races, religions, and nationalities. Each person who participated in this study had different barriers with regard to dealing with dying patients. Moreover, during the interviews, the participants described the barriers in depth; they provided the researcher with valuable information about their experience regarding critical situations to help the researcher to obtain significant results for the study. The barriers faced by novice nurses in Saudi Arabia, when working with dying or the sudden death of a patient, were identified and classified into four themes: (1) lack of experience with dying patients; (2) language barrier with patients and medical staff; (3) inadequate staffing, and (4) patients’ cooperation. The facilitators were placed into three themes: (1) caring/compassion, (2) teamwork/collaboration, and (3) mentored/experienced colleagues (Table 3).

### 3.1. Barriers

#### 3.1.1. Lack of Experience with Dying Patients

The participants felt that there were limited opportunities in clinical training to care for dying patients, so that, when they started working in the hospital, they were not ready to be involved with dying patients due to a lack of experience with these situations. The participants reported that their nursing program did not provide sufficient clinical hours. After graduation, they did receive a year of training as an intern; however, the internship program was not structured, so they did not receive enough one-on-one mentoring. The medical team did not want to be responsible for any mistakes made by the student. As described by one nurse, “…. I just wished that I had more experience in dealing with these situations, because during my bachelor’s degree, I was not exposed to many real-life situations, and our education does not focus on the practical side of nursing. In other words, our educational system does not make us 100% ready to practice nursing after graduation”. Another nurse commented that, “I did not have much experience dealing with critical patients”. The lack of guidance led to the students not receiving a proper education. This barrier could pose a serious threat to new nurses’ performance, if they were not well prepared before caring for patients.

#### 3.1.2. Language Barrier with Patients and Medical Staff

A significant barrier faced by all the participants was language. Nurses and medical teams in Saudi Arabia are required to communicate in English, but this was not the issue because all medical providers speak English fluently. However, Saudi Arabia has a significant level of diversity among both patients and medical teams, so there was an issue in communicating among them due to the different languages and nationalities; moreover, initially, their native language was not English. It is challenging to communicate with foreign patients correctly due to the patients’ languages; they do not speak Arabic or English, so the nurses do not know whether they are providing the best care for the patients or not. In addition, there are some who speak English, but sometimes the nurses cannot understand what the patient is saying due to their accent. Moreover, patients usually have different religions, beliefs, and cultures; so, if the nurse cannot understand the patients’ language, he or she would not be successful as a nurse.

Furthermore, most patients in Saudi Arabia speak Arabic as their first language, and they do not understand English, which makes communicating between patients and foreign nurses difficult. Caring for patients who speak a language in which the medical teams are not proficient in can decrease the quality of health services in Saudi Arabia. One participant commented in regard to the language barrier and stated, “Foreign language barrier because I am Filipino, and I can’t speak well in Arabic, so I can’t understand what the things that my patients are saying mean”. Another nurse stated, “We have foreign people who do not speak Arabic or English, and we also have many different accents that are hard to understand. That plays a vital role in affecting our communication with patients”.

Lastly, the language barrier is not only limited to the patients, but also affects the medical teams, who sometimes have difficulties in understanding each other due to the different dialects. The medical teams in the hospital usually have diverse employees from different countries—such as the Philippines, Egypt, India, Pakistan, and Saudi Arabia—and everyone speaks English in a different accent. One participant stated that “during critical situations…there are some…sometimes it’s hard for us to understand each other because all the people are in stress, so they cannot convey well what they are trying to say so there will be a communication problem”. This barrier can make the process of caring for patients difficult and incomplete. As described by another nurse, “one of the major issues we have in the hospital is that we must deal with a diverse population which made the communication difficult with patients, especially in critical situations”.

#### 3.1.3. Inadequate Staffing

Another barrier found from the interviews was the shortage of medical staff in the hospital. Most of the sample agreed that a lack of medical teams is one of the most challenging aspects that they face when working with critical situations. Moreover, the shortage of medical teams in the hospital made the process of caring for dying patients quite challenging, especially with the high influx of patients. Thus, lacking staff can affect the caregivers in providing holistic and complete care for everyone. As stated by one nurse,

“…Shortage in the medical teams made it hard to provide full attention to the dying patients. Collaboration in the nursing environment means covering shifts, assisting each other, and training new staff to be professionals. However, I believe that the reason for the shortage of medical teams in Saudi Arabia is free health care and that increases the number of patients over the medical teams…”

The shortage of medical teams in hospitals can affect the welfare of dying patients, leading to their premature death. Moreover, hospitals with inadequate staffing can exert a negative impact on nurses’ behavior during the process of caring for others, and nurses cannot perform their job if they are in an exhausting work environment. One participant pointed this out:

“…of course, we have a lack of staff, we cannot deliver sometimes…doctors’ orders will be delayed because we are with other patients helping the most critical situation first which is going to affect the other patients and not just the medications, but also the routine which is the laboratories, the ECGs are also delayed because of the lack of staff…”.

Nurses in the hospital cannot cover all patients’ needs due to the deficits in medical teams and supplies there. The approached hospital is considered one of the largest hospitals in Saudi Arabia, and it requires enough staff to cover all patients’ needs perfectly to improve the quality of healthcare. Moreover, the participants believed that the lack of hospital staff played an important role in affecting nurses’ performance and rendering them exhausted from working with patients, because they cannot bridge the shortage gap alone if the hospital does not have enough workers.

#### 3.1.4. Patients’ Responses or Cooperation

The final barrier was found to be the most challenging aspect to deal with, which was the beginning of a treatment, because there is a long process involved in diagnosing the patients’ situations accurately, especially with dying patients. Moreover, it was difficult for the participants to ensure that the patient would be able to collaborate with them in order to facilitate the treatment and to not place the nurses under pressure when they perform their job. Moreover, when patients arrive at the hospital, it is usually difficult to obtain enough information from them, especially with dying patients’ situations, due to their conditions during the treatment, and this may affect its quality. New nurses in Saudi Arabia tend to be afraid and nervous when they are involved with dying patients, especially with those who are unconscious. They believe that they cannot ensure that they are choosing the best treatment to protect the patient or not. One nurse commented that, “It is always challenging at the start of the treatment…. in order to move critical patient status from dangerous positions to a stable situation…” This barrier can negatively affect the nurses’ performance if they are not well prepared, and it may affect the trust between the patient and the medical team.

Additionally, new Saudi nurses face huge pressure and fear when they deal with a car accident patient, because they refuse to be the reason for the patient’s death. Usually, nurses feel more emotional towards the patients and this may affect the quality of the treatment. The most challenging part was described by one nurse as follows: “Those patients who have just come out from a horrible car accident specially when they have a lot of broken organs...it is really hard to work with them as it is emotionally challenging…” In addition to this, the patients’ families or companions sometimes cannot imagine that they may lose their patient, so they may interrupt and distract the healthcare givers by asking many questions about their patients’ status, which plays an important role in decreasing the medical team’s focus on the situation.

Furthermore, some families and companions tend to blame the medical for the outcome of the situation and this can cause a negative impact on the medical team’s behavior. Therefore, the quality of healthcare will decrease due to the huge pressure in avoiding any mistakes towards the patient. As described by one participant, “when the patient and his or her family are convinced that we are doing everything possible to help the patient, but we cannot control death, it makes it easier for us, because sometimes they blame nurses and doctors”. Finally, patients’ responses or cooperation with medical teams play an important role in changing the process of caring from the nurses’ side, and this may affect their passion for helping others negatively if patients do not show any improvement after receiving the treatment.

### 3.2. Facilitators

Facilitators are assisting factors that help to overcome barriers faced by individuals. The facilitators that novice nurses in Saudi Arabia have encountered during the process of caring for dying patients were identified in terms of three themes: (1) caring/compassion, (2) teamwork/collaboration, and (3) mentors/experienced colleagues. This section will describe the themes in detail.

#### 3.2.1. Caring/Compassion

A passion to help and care for people in need is the most common facilitator that the participants reported. The participants had expressed a passion for helping others since they were young, which helped them to decide on becoming nurses as a suitable career for them. Moreover, they believed that a passion for helping others is essential in the nursing field to ensure that nurses are creative and successful in their job. One participant commented, “I have chosen to be a nurse because I love to care for other people”, which played an important role in facilitating the process of caring in this nurse. Moreover, each participant expressed their feelings towards helping patients in a different way, such as the following comment by another participant: “I decided to be a nurse because I like taking care of the patient who needs our help as professional.” The “caring/compassionate” element of their work helped the participants manage their roles as professional nurses, as well as encouraging them to work hard to satisfy all the patients’ needs accurately—especially with dying patients. The participants believed that if they were not able to “control their fear of death”, they would not be able to handle a large amount of pressure, thus not deserving to be called nurses. Caring and compassion enabled the participants to overcome any barrier that could affect their performance. As discussed by one nurse, “since I was a kid, I had a passion towards helping people who are in need whether their need was medical or something else, so I found nursing the perfect career for me”. Although most nurses tend to be creative and skillful in their workplace, they sometimes need motivation to increase their efforts, so the participants used their compassion for others to provide the best care for their patients. Caring/compassion was considered as a great facilitator that helped them to manage their roles as professionals.

#### 3.2.2. Teamwork/Collaboration

Good collaboration between medical teams is believed to be an effective strategy to facilitate the process of caring for all patients. All participants agreed that the hospital environment is a healthy one for nurses to be successful, and their management is helpful for new staff to improve their performance. Moreover, medical teams in the hospital work collaboratively with others, to cover all patients’ needs adequately. One nurse commented that “Kindness and good collaboration between colleagues is the most positive advantage in our hospital which is the best facilitator to improve our ability to receive and protect the patients…”. Teamwork/collaboration is another effective facilitator that helped the participants to become successful in their workplace.

Moreover, they believed that in order to provide the best care for all patients, good collaboration between medical teams is essential to meet all patients’ needs accurately and to improve the healthcare society further. One participant pointed out, “I think collaboration is an amazing thing and it is essential in our career, because we need to save time sometimes, but at the same time, accuracy is a must which can only be done by teamwork”. Furthermore, the participants believed that the role of nurses is to protect, help, and diminish patients’ pain. Therefore, this cannot be achieved unless they work together as a team to accomplish this goal. As stated by one nurse, “fortunately, our hospital has a very healthy environment and our management is always trying to improve our facility more and more…which helps us respond to patient needs…”

#### 3.2.3. Mentors/Experienced Colleagues

The support of friends, administrators, colleagues, and experienced nurses was found as one of the most potent and effective ways to facilitate the process of caring, which helped the participants to overcome challenges during their work. The subject nurses in this sample stated that whenever they experienced difficulties in dealing with certain situations, they tended to seek help from their superiors to learn and improve their performance: “Our mentors and experienced colleagues were the greatest facilitators for us to improve our skills and increase our confidence and this helped to enhance the quality of health care in the hospital…”

As stated by another participant, “…we have mentors as well as experienced colleagues, and we are always welcome to discuss our feelings with them or any difficulty that we face. Besides, if a nurse’s performance drops, he gets called by his mentor to explain the reasons whether they are related to feelings or something else…”

Having this support from mentors and experienced colleagues helped the participants to manage their roles and be successful in their workplace. Moreover, this support enhanced the participants’ experience in dealing with barriers associated with critical situations. One participant commented, “When I asked for help, my colleague did not hesitate to help me and educated me while treating the patient”. This played an important role in developing the participants’ performance and made them ready to face any situation. As stated by one participant.

“…I relied on my colleague to help me to control cases. However, over time, I became an expert because I have learned from my colleague a lot such as I am in a place that requires me to save people not to be the reason of making their situations more complicated. That encouraged me to improve my confidence when I treat patients…”

## 4. Discussion

This study focused on the barriers and facilitators faced by novice nurses in Saudi Arabia, when working with the dying or sudden death of a patient, in order to create solutions for the barriers that could help to prepare new nurses to face any situation. The participants provided the most common barriers that they faced as new nurses. Four barriers impacted the performance of the participants at that time. All these barriers played an essential role in decreasing the participants’ confidence. This led to the novice nurses becoming afraid and nervous of caring for dying patients for the first time.

The first barrier was the lack of experience with dying patients. This is consistent with previous research in which nursing staff at regular wards, neurological units, and critical care units were assessed by Kostka et al. [7]. The mortality of a patient was shown to be the most distressing aspect for those operating in the neurological unit; however, this same event was found by the employees of the critical care unit not to be as distressing as resuscitation. The research by Üstükuş and Eskimez [6] also confirms the significant degree of anxiety amongst nurses. Younger and undertrained workers are the most prone to anxiety, according to a study conducted in 2000. For nursing staff who consider mortality to be one of the most challenging clinical situations, caring for dying patients and assisting their families is a painful and demanding duty. Nurse practitioners must possess the proper information, abilities, and a clear method to handle a client in order to be capable of coming to terms with such scenarios. Acceptance, proficiency, emotional diversion, and expertise, as seen in Filipino nurses, are equally critical in coping with a patient’s mortality.

The second barrier is the multiple languages that patients speak. The language barrier was found to be the most common barrier that the participants faced during the process of caring. This is consistent with previous research in Singapore, which is home to a diverse international population. It has also evolved into a center for foreign patients of many cultural and religious origins. These circumstances make it difficult for nurses to interact successfully with clients [18]. Another challenge is the growing number of international nursing professionals, who are finding it more challenging to communicate with members of Singapore’s different cultures [19]. Interactions cannot be successful if there is no common language, and even body language might be misconstrued between cultures. Clients are also less responsive to nurses with whom they do not speak the same language or have a similar culture [20].

The next barrier is inadequate staffing of the facility. The nurses’ ability to directly care for the assigned patients suffered from a lack of help from others. This is in line with recent research, which indicated that nurses were frequently unable to devote adequate time or offer continuation of care to dying individuals and their relatives due to the conflicting requirements of severely ill patients and staffing shortages [21]. This requirement for extra time has also been documented in previous studies looking at the administration of end-of-life care in the emergency department in connection to a shortage of skilled nurses in the United States [22].

The final barrier that the participants faced during the process of caring for dying patients was the patients’ responses to or cooperation with the treatment plan. This barrier affected the participants emotionally, which impeded their growth as professional nurses. This is consistent with a previous study by Cicolello et al. [23], which found that dying patients expressed their mistrust of the healthcare system, where they refused to obtain care and rejected treatment at many instances, which was also enhanced by a lack of cooperation from family members.

## 5. Conclusions

Nurses face a major challenge in dealing with dying patients, and these situations can pose a serious threat to nurses’ behavior if they do not prepare themselves well. Moreover, it is important to identify the factors that may have an impact on novice nurses’ performance while dealing with dying patients and their families in order to prepare an action plan that can improve their performance and offer high-quality healthcare to patients. Furthermore, nurses who are more experienced in assisting the families of patients with no hope of a cure, given their pain and suffering, should engage in open discussions with less experienced and newly graduated nurses who are about to begin this type of practice. An essential obstacle is determining how this teaching process should be implemented, and how we can ease the transition for less experienced nurses. Finally, more research is needed to identify the difficulties faced by new nurses while dealing with dying patients, because this topic is crucial as it involves people’s lives.

## Figures and Tables

**Table 1 healthcare-10-02259-t001:** Interview questions.

**Introduction Question**	How do you feel about the term “novice nurse”?
**Transition Question**	What do you think of death in general?
**Key Questions**	How do you perceive the death of a patient?
	How do you describe the factors that help deliver care for a dying patient?
	How do you describe the factors that hinder you from delivering care for a dying patient?
	How do you describe the interactions with the family of a dying patient?
**Final prompt**	Do you have anything further to say?
**Probing Questions**	Could you give us a better description?
	Could you provide us a better description?

**Table 2 healthcare-10-02259-t002:** Participant characteristics.

Variable	Category	*N*	*%*
**Gender**	Male	8	66.6
	Female	4	33.3
**Nationality**	Saudi	5	41.6
	Filipino	7	58.3
**Age (M ± SD)**	20–25	4	33.3
	26–30	5	41.6
	31–35	3	25.0

**Table 3 healthcare-10-02259-t003:** Coding tree.

Main Themes	Subthemes	Verbatim Excerpts
Barriers	Lack of experience with dying patients	…I just wished that I had more experience… …I did not have much experience dealing with critical patients…
	Language barrier with patients and medical staff	…Foreign language barrier because I am Filipino… …Sometimes it is hard for us to understand each other…
	Inadequate staffing	…Shortage in the medical teams made it hard…
	Patient’s cooperation	…It is really hard to work with them…
Facilitators	Caring/compassion	…I like taking care of the patient who needs our help…
	Teamwork/collaboration	…Kindness and good collaboration between colleagues…
	Mentored/experienced colleagues	…We have mentors as well as experienced colleagues…

## Data Availability

The data will be shared by the authors of this research paper upon request.

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
