# Peer review of "Barriers and Facilitators of Working with Dying Patients among Novice Nurses in Saudi Arabia"

_healthcare, 2022, doi:10.3390/healthcare10112259_

Round 1

Reviewer 1 Report

This is a qualitative study that aimed at identifying the barriers and facilitators of working with dying patients among novice nurses in Saudi Arabia.

The topic is interesting and relevant to the clinical nursing practice; however, the overall quality of the writing is very poor. 

Line 21 Abstract: what do you mean by this "let alone new nurses"? 

Line 35-48: please delete this irrelevant part

Line 86: "Therefore, the aim of the study was to evaluate the perceptions of Saudi novice nurses regarding death of patients." this is inaccurate compared to the aim in the abstract 

please improve the quality of writing and I'm willing to re-review 

Author Response

Dear reviewer, thank you so much for all the efforts that you have exerted in reviewing our paper. We have sent the article to an English language expert for thorough editing. As for the specific comments provided by you we have changed “let alone new nurses” in 21 to read “especially new nurses”. And we have deleted the section from line 35 to line 48 as warranted. As for the aim of the study we have unified it to read, “this study aimed at exploring the barriers and facilitators of working with dying patients experienced by novice nurses in Saudi Arabia”.

Reviewer 2 Report

The article is valuable, it can be used to update the educational programs during the student period in clinical fields, and it will provide good information to health system managers.

Phenomenological research aims to explain and identify phenomena as people perceive them in a specific situation.Phenomenological paradigms are classified as interpretative paradigms based on the phenomenological paradigm classification.It refers to a scientific method that summarizes social issues using a mental approach.

Based on the phenomenological method used in this article, it is important to extract the original experiences of the participants; In this article, the phenomenology method is not implemented correctly and the directional content analysis method is used instead.The working method of this article is inappropriate for the type of research, so it should be fundamentally revised and classes and sub-classes extracted using the phenomenology method. 

The extracted classes and sub-classes are currently written using the content analysis method, which should be modified or changed to directional content analysis.

Author Response

Dear reviewer, thank you so much for your genuine interest and valuable input on our work. We have used phenomenology in this research study rather than content analysis as our aim was to explore the lived experiences of novice nurses in caring for dying patients and through these experiences, identify the opportunities and challenges that they perceive in the context of their work. We do understand and appreciate your point of view and we believe that maybe referring to the barriers and facilitators with terms such as “identify the variables” might have gave that impression, therefore, we have adjusted the aim of the study to read “ this study aimed at exploring the barriers and facilitators of working with dying patients experienced by novice nurses in Saudi Arabia”. Another issue that might have gave the impression of use of content analysis was the following phrase, “The data was analyzed using a conventional qualitative content evaluation technique.” Unfortunately, this was a mistake in writing and reporting rather than a true reflection of the work which has been put into design the phenomenological study, which included the use of Coliazzi’s phenomenological approach, therefore the following statement was deleted, “The data was analyzed using a conventional qualitative content evaluation technique. We listened to the sections of the conversation that were relevant to the participants' encounters”.  

Reviewer 3 Report

Dear author,

Thank you for the opportunity to review the paper.

This study focused on the barriers and facilitators faced by novice nurses in Saudi Arabia when working with dying or sudden death. 

I think the authors did a very good effort to write a sound scientific paper with relevant references through the described study but were based on a too-limited and not homogeneous study sample. 

I suggest adding these points: 

Colaizzi's Method follows seven data analysis steps. Could you please improve this?

I suggest adding a completed COREQ list for the reporting of your qualitative research process

I suggest adding table 3 with information that was identified and organized into themes or categories.

I hope my comments may be of help to authors in their work.

Author Response

Dear reviewer we immensely appreciate your effort in reviewing our paper and in providing us with this valuable insight. Regarding the sample we tried to achieve diversity within the practice setting considering the diverse nationalities of the nursing staff and mainly novice nurses working in Saudi hospital who work within the framework of a unified practice legislation. Therefore by using maximum variation sampling, we attempt at achieving heterogeneity within the homogeneity of the sample represented by the fact that all the participants recruited were novice nurses working at one ER department of a major hospital in Saudi Arabia. This would help in identifying the shared experiences of these nurses across the diversity.

Regarding the Colaizzi method we have improved it as such.  

As per the Colaizzi approach, the researchers read the data repeatedly, extracting significant data, coding recurring data, and lastly organizing the data as themes. The Colaizzi method consists of seven steps: (1) Read the transcript to be familiar with the data; (2) Identify and extract significant statements and phrases; (3) Formulate meanings; (4) Group all formulated meanings into categories, clusters of themes and themes; (5) Define all emergent themes into an exhaustive description; (6) Describe the fundamental concepts of the phenomena; and (7) approach participants for further information [14].

We have also added the coding tree and attached a COREQ checklist.

Round 2

Reviewer 1 Report

No further commnets 

Reviewer 3 Report

Dear Authors,

I appreciate the work that went into this study. I thank the authors for the thoroughness of their responses, and I recommend acceptance as is.

Best regards